# Evaluating the content validity of two versions of an instrument used in measuring pediatric pain knowledge and attitudes in the Ghanaian context

Abigail Kusi Amponsah[1,2☝]*, Victoria Bam[2‡], Minna Stolt[1‡], Joonas Korhonen[1‡], Anna Axelin[1‡]

1 Department of Nursing Sciences, Faculty of Medicine, University of Turku, Turku, Finland, 2 Department of Nursing, Faculty of Allied Health Sciences, Kwame Nkrumah University of Science and Technology, Kumasi, Ghana

☝ These authors contributed equally to this work.
‡ VB, MS, JK and AA also contributed equally to this work.
* abkuam@utu.fi, akamponsah.fahs@knust.edu.gh, abacious@live.co.uk

**Data Availability Statement:** All relevant data are within the manuscript and its Supporting Information files.

## Abstract

In this article, we compared the content validity of two instruments used in measuring pediatric pain knowledge and attitudes. This was considered necessary due to the universal differences in culture, semantics and healthcare resources in different parts of the globe. Thirteen (13) pediatric experts in Ghana assessed the content validity of two instruments: the 42-item Pediatric Nurses' Knowledge and Attitudes Survey Regarding Pain (PNKAS) and the 41-item Pediatric Healthcare Providers' Knowledge and Attitudes Survey Regarding Pain (PHPKASRP). The relevance and clarity of each item on these instruments were rated on a four-point likert scaled options from 1 (not relevant/ not clear) to 4 (very relevant/ very clear). The item-level content validity index (I-CVI) was calculated by dividing the number of experts who rated an item with 3 or 4 by the total number of experts. The average scale-level content validity index (S-CVI/Ave) was also estimated by summing up the I-CVIs of all items and dividing them by the total number of items. The I-CVIs on the PNKAS ranged from 0.62 to 1.00 for the relevance component and 0.69 to 1.00 for the clarity component. The I-CVIs on the PHPKASRP ranged from 0.62 to 1.00 for both the relevance and clarity components. The S-CVI/Ave were 0.87 and 0.89 for the relevance and clarity aspects on the PNKAS respectively. The S-CVI/Ave for the PHPKASRP instrument were 0.86 and 0.89 for the relevance and clarity aspects correspondingly. At the end of the validation process, 5 items were revised on both instruments whilst 37 and 36 items were maintained on the PNKAS and PHPKASRP instruments respectively. The PNKAS and PHPKASRP have an acceptable level of content validity in the Ghanaian context and recommended for educational and research purposes. Other forms of validity and reliability of these instruments should also be examined in future studies.

**Funding:** The authors received no specific funding for this work.

**Competing interests:** The authors have declared that no competing interests exist.

## Introduction

Unrelieved pediatric pain remains a significant health problem globally despite decades of research on pain and advanced technologies [1–3]. Inadequately treated pain does not only affect the child-in-pain but also their family and society at large. Earlier studies have demonstrated the untoward consequences of untreated or inadequately treated pain on children to include impaired physical function, emotional disturbances [4], social isolation [5], delayed recovery, prolonged hospitalization, increased cost of healthcare, and development of chronic pain which decreases their quality of life [6–8]. The socio-economic burden of this menace on families and societies has also been documented in the pain literature [9, 10].

Nurses and other healthcare providers (HCPs) play a significant role in the assessment and management of hospitalized children's pain [11]. The assessment of HCPs' knowledge and attitudes regarding children's pain is important as it serves as the bedrock of their pain assessment and management practices in this vulnerable population [12]. Considering the importance of assessing HCPs' knowledge and attitudes regarding children's pain, a number of instruments have been developed to measure this construct [13, 14]. Key among them is the Pediatric Nurses' Knowledge and Attitudes Survey regarding pain (PNKAS) [13]. According to the developer (Manworren, R. C. B. via email communication), the PNKAS was revised in 2014 to reflect the diverse roles performed by the multidisciplinary healthcare team involved in providing pain care for pediatric patients; this occasioned the development of the Pediatric Healthcare Providers' Knowledge and Attitudes Survey Regarding Pain (PHPKASRP).

Following the development of the PNKAS instrument, its validity and reliability has been assessed [13]. Face and content validity of the instrument was established by five pain management experts in the United States of America. Internal consistency of the PNKAS instrument has also been assessed by two distinct groups with a reported acceptable Cronbach's alpha value of 0.72 among 247 pediatric nurses and 0.77 among 88 members of a children's nursing organization. Test-retest reliability analysis among 12 clinicians (6 nurses and 6 child life specialists) recorded a correlation coefficient of 0.67, signifying an acceptable level of instrument stability. The PNKAS instrument has been modified to suit nurses taking care of children who do not have any form of malignancies; this version has been termed as the Modified Mongolian Pediatric Nurses' Knowledge and Attitudes Survey-Shriner's version (MMPNKAS-S) [15]. The instrument has also been translated into the Norwegian language and has demonstrated an acceptable linguistic validity [16].

In lieu of diverse roles performed by the multidisciplinary pediatric pain team, the revised version (PHPKASRP) became necessary. Presently, there are no published findings on the psychometric properties of the PHPKASRP. In spite of this, the instrument has been used to assess changes in healthcare providers' knowledge and attitudes following a multidisciplinary educational intervention program [17].

Validity is not a property of an instrument but dependent on the interpretation or purpose of an instrument with particular context and participants [18] due to universal differences in culture, semantics and resources in different parts of the globe [19, 20]. It appears from the review of relevant literature that, the content validity of these two instruments have not been assessed from a low-middle income country's context. As part of plans to use a pediatric pain knowledge and attitude instrument as a tool in assessing the effectiveness of a short-course educational program for nursing students and nurses, the current study sought to evaluate and compare the content validity of both the original PNKAS instrument and its revised version, PHPKASRP from a Ghanaian perspective.

## Development of the PNKAS and PHPKASRP instruments

The Pediatric Nurses' Knowledge and Attitudes Survey Regarding Pain (PNKAS) is a modification of Nurses' Knowledge and Attitudes Survey Regarding Pain (NKASRP) instrument developed by Ferrell and McCaffery in 1987 [13]. As the name implies, it was originally developed to measure nurses' knowledge and attitudes toward patients in pain [21]. The content of the NKASRP tool was derived from the prevailing standards of pain management from organizational bodies such as the American Pain Society, the World Health Organization, and the National Comprehensive Cancer Network Pain Guidelines. The tool since its development has undergone revisions to reflect changes in pain management. The NKASRP consists of 39 items and take about 25–30 minutes to be completed. The instrument has been proven to be valid and reliable in different settings [22, 23].

In order to develop a tool which would be more sensitive to pediatric patients, Manworren developed a new survey in 1998 called the "Pediatric Nurses' Knowledge and Attitudes Survey Regarding Pain" (PNKAS) based on the original work of McCaffery and Ferrell [24]. Three major changes were executed by Manworren in order to change the focus of the NKASRP tool from adults to infants, children and adolescents. The first amendment was adding three procedural pain items (questions 8, 14, and 21) to the original survey. Secondly, she modified questions that were related to meperidine and aspirin to other recommended analgesics due to the effects of these analgesics on the pediatric population. Aspirin increases the risk of Reye's syndrome [25], whereas meperidine has toxic metabolic effects [26], hence, they are contraindicated in children's pain management [13]. Thirdly, the dosage of analgesics in the original questionnaire was adjusted to suit paediatric patients. For instance, morphine dosages (in question number 26), were adjusted to fit the paediatric population. The 42-item PNKAS is a self-administered instrument that assesses nurses' knowledge and attitudes regarding pain assessment and management in the pediatric population [24]. It comprises of 25 binary response-type questions (True/ False), 13 multiple choice questions (MCQs), and two case studies extended into four MCQs.

The Pediatric Healthcare Providers' Knowledge and Attitudes Survey Regarding Pain (PHPKASRP) is a revised instrument developed from the Pediatric Nurses' Knowledge and Attitudes Survey Regarding Pain (PNKAS). In 2014, the 41-item self-administered revised version (PHPKASRP instrument) was developed to reflect the different roles of the multidisciplinary healthcare team involved in providing pain care for pediatric patients. The revised instrument which takes about 25–30 minutes to complete consists of 25 binary response-type questions (True/ False), 12 multiple choice questions (MCQs), and two case studies extended into four MCQs.

## Materials and methods

### Study design, sample and setting

This descriptive methodological study is part of a larger project that sought to examine the educational needs on pediatric pain management in Ghana. Based on the recommended minimum number of eight to 12 experts [27], the researchers physically contacted 15 pediatric experts to assess the content validity of both the PNKAS and PHPKASRP instruments from a Ghanaian perspective. The experts were given hard copies of these instruments and had to complete them within two to four weeks before returning them to the researchers. The pediatric experts were from eight hospitals and four nursing educational institutions located in the Ashanti region of Ghana; and were contacted between October 2018 and February 2019. The experts were chosen based on their level of training, clinical and/ or teaching experience in pediatrics.

The pediatric care settings in Ghanaian healthcare facilities take care of sick children with medical and surgical conditions on both out-patient and in-patient basis. The in-patient pediatric care settings consist of incubators, cots and beds which accommodate children from birth up to 12 or 13 years old depending on the facility's protocol. The pediatric settings are sub-divided into various sections based on the procedures performed, children's age or severity of their condition (medical or surgical). The units are colourfully painted with child-friendly designs and have television sets and toys for entertainment purposes; there are also designated areas with resources for playing purposes. Sick children are accommodated with at least one family caregiver or guardian during hospital admissions. Healthcare is mainly provided to sick children and their families by physicians, nurses, pharmacists, dietitians, physiotherapists, psychologists, healthcare assistants among others. Vital signs monitoring (including pain as the fifth vital sign) form an integral part of the role of healthcare providers towards hospitalised children and their families.

## Data collection instrument, procedures and analysis

The pediatric experts rated the relevance and clarity of both instruments (the 42-item PNKAS instrument and the 41-item PHPKASRP instrument) on a four-point rating scale with 1 (not relevant or not clear), 2 (somewhat relevant or somewhat clear), 3 (quite relevant or quite clear) or 4 (very relevant or very clear) as done in earlier studies [28, 29]. In addition, they were required to make comments on each of the items regarding their grammatical construction, simplicity, representativeness, comprehension, ambiguity, modification (deletion or addition) among others as they deemed appropriate [30].

Both individual item-level and scale level content validity indices were estimated for both relevance and clarity aspects of the two instruments (PNKAS and PHPKASRP). Item-level content validity index (I-CVI) was calculated by dividing the number of experts who rated an item with a score of 3 (quite relevant/ quite clear) or 4 (very relevant/ very clear) over the total number of experts [31, 32]. As a general criterion, I-CVI should be ≥ 0.70 [33] to be regarded as measuring an appropriate sample of the instrument items for a particular construct. In line with the recommendations of Delgado-Rico and colleagues [34], decisions on items (i.e., elimination, modification or conservation) were made on the basis of the content validity indices, feedback given by experts, inputs from the instrument developer and the contribution of the items to the overall construct under investigation. The scale-level content validity index (S-CVI) or average scale-level content validity index (S-CVI/Ave) focuses on the average item quality and is estimated by summing up the I-CVIs of all items and dividing them by the total number of items [29, 32, 35]. The minimum acceptable value of S-CVI/Ave should be 0.80 [31, 32, 36]; values greater than or equal to 0.90 are considered as excellent average [37].

The pediatric experts' comments on the individual items of the instruments were first reviewed by the research team. On the basis of the nature of the comment, the research team made decisions regarding the elimination, modification or conservation of the items involved. The pediatric experts' comments and the research team's decisions were then sent to the instrument developer for her review and feedback. Final decisions on the items were made through dialogue between the instrument developer and the research team. These decisions were also underpinned by the prevailing research evidence and the contribution of each item to the overall goals of the instruments.

The data were initially entered and cleaned in Microsoft Excel before being exported into Statistical Packages for the Social Sciences (SPSS) version 25.0 (SPSS, Chicago, IL, USA) for further analysis. Frequencies of items which were rated as relevant/ irrelevant and clear/ unclear were calculated. The proportion or percentage of items which were rated as relevant

and clear by the pediatric experts were also estimated. Additionally, the mean I-CVIs of both the 42-item PNKAS instrument and the 41-item PHPKASRP instrument were determined.

## Ethical considerations

Ethical approval for the study with reference number CHRPE/AP/574/18 was provided by the Committee on Human Research, Publications and Ethics (CHRPE), School of Medical Sciences (SMS), Kwame Nkrumah University of Science and Technology (KNUST), Ghana. Participants in the present study signed a written informed consent form and submitted the data collection instrument after completion. They were assured of anonymity, confidentiality, and their right to voluntary participation in the study. Authorization for the use of the PNKAS and PHPKASRP instruments in the current study was granted by the instrument developer, Manworren on August 16, 2018 (via email communication).

## Results

### Demographic characteristics of the pediatric experts

Thirteen (13) out of the 15 pediatric experts completed and returned the data collection instrument, yielding a response rate of 87%. The experts comprised of four pediatric nursing educators, eight pediatric nurses and one pediatrician. Their median (range) age was 38 (32–51) years (refer to Table 1). Majority of them were female (69%) and had a postgraduate degree (61.5%). The experts had worked in the healthcare profession for a median duration of 13 years and in pediatrics for 6 years.

### Content validity assessment of the PNKAS instrument

The number of items considered relevant (with a rating of 3 or 4) by all 13 pediatric experts was five (refer to Table 2). The I-CVI for the 42 items ranged from 0.62 to 1.00 for the relevance aspect of the instrument. Four (4) out of the 42 items fell below the recommended I-CVI of 0.70; thus, the proportion of items considered relevant on this basis was 90.5%. The average relevance CVI for the scale (S-CVI/Ave) was 0.87, indicating an acceptable level of content validity and slightly below the 0.90 accepted excellent value.

The number of items considered clear (with a rating of 3 or 4) by all the 13 experts was 13. The I-CVI for the 42 items ranged from 0.69 to 1.00 for the clarity component of the PNKAS instrument. Two (2) out of the 42 items fell below the recommended I-CVI of 0.70; the proportion of items considered clear on this foundation was 95.2%. The average clarity CVI for the

**Table 1. Demographic characteristics of pediatric experts (n = 13).**

| Variables | Frequency (%) | Median (range) |
|---|---|---|
| *Age (years)* | | 38 (32–51) |
| *Gender* | | |
| Male | 4 (30.8) | |
| Female | 9 (69.2) | |
| *Working years in the health profession* | | 13 (7–24) |
| *Working years in pediatrics* | | 6 (3–14) |
| *Educational level* | | |
| Bachelor's degree | 5 (38.5) | |
| Postgraduate degree | 8 (61.5) | |

**Table 2. Content validity assessments of PNKAS and PHPKASRP instruments by pediatric experts (n = 13).**

| Items on the PNKAS 1999 version (*Answer*) | PNKAS | | Items on the PHPKASRP 2014 revised version (*Answer*) | PHPKASRP | | Pediatric Experts' Comments (Number of Experts) | Action Taken; Revised Form |
|---|---|---|---|---|---|---|---|
| | Relevance; I-CVIs | Clarity; I-CVIs | | Relevance; I-CVIs | Clarity; I-CVIs | | |
| Q1_Observable changes in vital signs must be relied upon to verify a child's/ adolescent's statement that he/ she has severe pain. (*False*) | 13; 1.00 | 13; 1.00 | Q1_Observable changes in vital signs must be relied upon to verify a child's/ adolescent's self-report of severe pain. (*False*) | 13; 1.00 | 13; 1.00 | This is quite broad, can question be directed at specific vital sign or signs e.g. heart rate (n = 1); Can do away with the / and use one of them (n = 1) | Kept; – |
| Q2_Because of an underdeveloped neurological system, children under 2 years of age have decreased pain sensitivity and limited memory of painful experiences. (*False*) | 12; 0.92 | 13; 1.00 | Q2_Because their nervous system is underdeveloped, children under 2 years of age have decreased pain sensitivity and limited memory of painful experiences. (*False*) | 12; 0.92 | 13; 1.00 | | Kept; – |
| Q3_If the infant/ child/ adolescent can be distracted from his/ her pain, this usually means that he is not experiencing a high level of pain. (*False*) | 10; 0.77 | 11; 0.85 | Q3_Pediatric patients (infants, children, adolescents) who can be distracted from pain usually do not have severe pain. (*False*) | 10; 0.77 | 11; 0.85 | Pediatric patients cover all so no need to put them all in brackets (n = 1) | Kept; – |
| Q4_Infants/ children/ adolescents may sleep in spite of severe pain. (*True*) | 12; 0.92 | 12; 0.92 | Q8_Infants/ children/ adolescents may sleep in spite of severe pain. (*True*) | 12; 0.92 | 12; 0.92 | | Kept; – |
| Q5_Comparable stimuli in different people produce the same intensity of pain. (*False*) | 11; 0.85 | 11; 0.85 | Q5_Comparable stimuli in different people produce the same intensity of pain. (*False*) | 11;0.85 | 11; 0.85 | | Kept; – |
| Q6_Ibuprofen and other nonsteroidal anti-inflammatory agents are NOT effective analgesics for bone pain caused by metastases. (*False*) | 11; 0.85 | 10; 0.77 | Q9_Ibuprofen and other nonsteroidal anti-inflammatory agents are NOT effective analgesics for pain from bone metastases. (*False*) | 11; 0.85 | 10; 0.77 | | |
| Q7_Non-drug interventions (e.g. heat, music, imagery etc.) are very effective for mild-moderate pain control but are rarely helpful for more severe pain. (*False*) | 12; 0.92 | 12; 0.92 | Q10_Non-drug interventions (e.g. guided imagery, biofeedback, transcutaneous electrical nerve stimulation (TENS) etc.) are very effective for mild-moderate pain control but are rarely helpful for more severe pain. (*False*) | 12; 0.92 | 12; 0.92 | Examples of the non-drug interventions are not familiar to practitioners in the Ghanaian context (n = 1); Examples should be contextualized or removed to generalize the question (n = 1). | **Amended**; Evidence-based non-drug interventions are very effective for mild-moderate pain control but are rarely helpful for more severe pain. (*False*) |
| Q8_Children who will require repeated painful procedures (e.g., daily blood draws), should receive maximum treatment for the pain and anxiety of the first procedure to minimize the development of anticipatory anxiety before subsequent procedures. (*True*) | 13; 1.00 | 11; 0.85 | Q6_Children who will require repeated painful procedures (e.g., daily blood draws), should receive maximum treatment for the pain and anxiety of the first procedure to minimize the development of anticipatory anxiety before subsequent procedures. (*True*) | 13; 1.00 | 11; 0.85 | Question should be simplified (n = 1) | Kept; – |

(*Continued*)

**Table 2.** (Continued)

| Items on the PNKAS 1999 version (*Answer*) | PNKAS | | Items on the PHPKASRP 2014 revised version (*Answer*) | PHPKASRP | | Pediatric Experts' Comments (Number of Experts) | Action Taken; Revised Form |
|---|---|---|---|---|---|---|---|
| | Relevance; I-CVIs | Clarity; I-CVIs | | Relevance; I-CVIs | Clarity; I-CVIs | | |
| Q9_Respiratory depression rarely occurs in children/ adolescents who have been receiving opioids over a period of months. (*True*) | 9; 0.69 | 11; 0.85 | Q7_Respiratory depression rarely occurs in children/ adolescents who have been receiving stable doses of opioids over a period of months. (*True*) | 9; 0.69 | 11; 0.85 | Not the common practice in Ghana (n = 1). | Kept; – |
| Q10_Acetaminophen 650 mg PO is approximately equal in analgesic effect to codeine 32 mg PO. (*True*) | 11; 0.85 | 11; 0.85 | | | | | |
| Q11_The World Health Organization (WHO) pain ladder suggests using single analgesic agents rather than combining classes of drugs (i.e. combining an opioid with a non-steroidal agent). (*False*) | 11; 0.85 | 10; 0.77 | Q11_Combining analgesics and non-drug therapies that work by different mechanisms (e.g. using acetaminophen, topical anesthetics, sucrose, and non-nutritive sucking) may result in better pain control with fewer side effects than using a single analgesic agent. (*True*) | 11; 0.85 | 10; 0.77 | Examples should be removed or positioned beside each intervention (n = 1). | **Amended;** Combining analgesics (e.g. using acetaminophen, topical anesthetics) and non-drug therapies (e.g. sucrose, and non-nutritive sucking) that work by different mechanisms may result in better pain control with fewer side effects than using a single analgesic agent. |
| Q12_The usual duration of analgesia of morphine IV is 4–5 hours. (*False*) | 12; 0.92 | 13; 1.00 | Q4_The usual duration of analgesia of morphine IV is 4–5 hours. (*False*) | 12; 0.92 | 13; 1.00 | This requires specific knowledge of morphine pharmacology (something I consider too detailed for basic nursing) (n = 1). | Kept; – |
| Q13_Research shows that promethazine (Phenergan®) is a reliable potentiator of opioid analgesics. (*False*) | 10; 0.77 | 13; 1.00 | Q12_Benzodiazepines do not reliably potentiate the analgesia of opioids' unless the pain is related to muscle spasms (*False*) | 10; 0.77 | 13; 1.00 | Another question that requires detailed knowledge (n = 1). | Kept; – |
| Q14_Parents should not be present during painful procedures (*False*) | 11; 0.85 | 12; 0.92 | Q13_Parents should not be present during painful procedures. (*False*) | 11; 0.85 | 12; 0.92 | | Kept; – |
| Q15_Adolescents with a history of substance abuse should not be given opioids for pain because they are at high risk for repeated addiction. (*False*) | 13; 1.00 | 13; 1.00 | Q14_Adolescents with a history of substance abuse should not be given opioids for pain because they are at high risk for repeated addiction. (*False*) | 13; 1.00 | 13; 1.00 | | Kept; – |
| Q16_Beyond a certain dosage of morphine, increases in dosage will NOT provide increased pain relief. (*False*) | 12; 0.92 | 13; 1.00 | Q15_Beyond a certain dosage of morphine, increases in dosage will NOT provide increased pain relief. (*False*) | 12; 0.92 | 13; 1.00 | | Kept; – |
| Q17_Young infants, less than 6 months of age, cannot tolerate opioids for pain relief. (*False*) | 10; 0.77 | 10; 0.77 | Q16_Young infants, less than 6 months of age, cannot tolerate opioids for pain relief. (*False*) | 10; 0.77 | 10; 0.77 | | Kept; – |
| Q18_The child/ adolescent with pain should be encouraged to endure as much pain as possible before resorting to a pain relief measure. (*False*) | 8; 0.62 | 11; 0.85 | Q18_The child/ adolescent with pain should be encouraged to endure as much pain as possible before resorting to an opioid for pain relief. (*False*) | 8; 0.62 | 11; 0.85 | | Kept; – |

(*Continued*)

**Table 2.** (Continued)

| Items on the PNKAS 1999 version (*Answer*) | PNKAS | | Items on the PHPKASRP 2014 revised version (*Answer*) | PHPKASRP | | Pediatric Experts' Comments (Number of Experts) | Action Taken; Revised Form |
|---|---|---|---|---|---|---|---|
| | Relevance; I-CVIs | Clarity; I-CVIs | | Relevance; I-CVIs | Clarity; I-CVIs | | |
| Q19_Children less than 8 years cannot reliably report pain intensity and, therefore, the nurse should rely on the parent's assessment of the child's pain intensity. (*False*) | 10; 0.77 | 13; 1.00 | Q19_Children less than 8 years cannot reliably report pain intensity and therefore, the healthcare provider should rely on the parent's assessment of the child's pain intensity. (*False*) | 10; 0.77 | 13; 1.00 | It does not apply to all children who are less than 8 years. (n = 1). | **Amended;** Most children as young as 4 years of age can reliably report pain intensity using a developmentally appropriate self-report tool. (*True*) |
| Q20_ Based on one's religious beliefs, a child/ adolescent may think that pain and suffering is necessary. (*True*) | 12; 0.92 | 13; 1.00 | Q17_Spiritual beliefs may lead a child /adolescent to think that pain and suffering are necessary. (*True*) | 12; 0.92 | 13; 1.00 | | Kept; – |
| Q21_Anxiolytics, sedatives and barbiturates are appropriate medications for the relief of pain during painful procedures. (*False*) | 9; 0.69 | 11; 0.85 | Q20_Anxiolytics, sedatives and barbiturates are appropriate medications for the relief of pain during painful procedures. (*False*) | 9; 0.69 | 11; 0.85 | | Kept; – |
| Q22_After the initial recommended dose of opioid analgesic, subsequent doses should be adjusted in accordance with the individual patient's response. (*True*) | 11; 0.85 | 13; 1.00 | Q21_After the initial dose of opioid analgesic is given, subsequent doses should be adjusted based on the individual patient's response. (*True*) | 11; 0.85 | 13; 1.00 | This may also require detailed knowledge beyond basic nursing (n = 1). | Kept; – |
| Q23_The child/ adolescent should be advised to use non-drug techniques alone rather than concurrently with pain medications. (*False*) | 10; 0.77 | 13; 1.00 | Q22_The child/ adolescent should be advised to use non-drug techniques alone rather than concurrently with pain medications. (*False*) | 10; 0.77 | 13; 1.00 | | Kept; – |
| Q24_Giving children/ adolescents sterile water by injection (placebo) is often a useful test to determine if the pain is real. (*False*) | 12; 0.92 | 12; 0.92 | Q23_Giving children/ adolescents sterile water by injection (placebo) is often a useful test to determine if the pain is real. (*False*) | 12; 0.92 | 12; 0.92 | | Kept; – |
| Q25_In order to be effective, heat and cold should be applied directly to the painful area. (*False*) | 11; 0.85 | 11; 0.85 | | | | | |
| | | | Q24_Sedation always precedes opioid related respiratory depression. (*True*) | 9; 0.69 | 8; 0.62 | I don't get this (n = 1); The use of "always" gives out the answer and makes it a leading question (n = 1). | Kept; – |
| Q26_The recommended route of administration of opioid analgesics to children with prolonged cancer-related pain is: (*oral*) | 12; 0.92 | 12; 0.92 | Q26_The recommended route of administration of opioid analgesics to children with prolonged cancer-related pain is: (*oral*) | 12; 0.92 | 12; 0.92 | | Kept; – |
| | | | Q27_The usual time to peak effects for traditional analgesics (acetaminophen, non-steroidal anti-inflammatory drugs, and opioids given orally is: (*60 minutes*) | 11; 0.85 | 11; 0.85 | | Kept; – |

(*Continued*)

**Table 2.** (Continued)

| Items on the PNKAS 1999 version (*Answer*) | PNKAS | | Items on the PHPKASRP 2014 revised version (*Answer*) | PHPKASRP | | Pediatric Experts' Comments (Number of Experts) | Action Taken; Revised Form |
|---|---|---|---|---|---|---|---|
| | Relevance; I-CVIs | Clarity; I-CVIs | | Relevance; I-CVIs | Clarity; I-CVIs | | |
| Q27_The recommended route of administration of opioid analgesics to children with brief, severe pain of sudden onset, e.g., trauma or postoperative pain is: *(intravenous)* | 12; 0.92 | 12; 0.92 | Q28_ The recommended route administration of opioid analgesics to children with brief, severe pain of sudden onset, e.g., trauma or postoperative pain, is: *(intravenous)* | 12; 0.92 | 12; 0.92 | | Kept; – |
| Q28_ Which of the following analgesic medications is considered the drug of choice for the treatment of prolonged moderate to severe pain for children with cancer? *(morphine)* | 12; 0.92 | 12; 0.92 | Q29_ Which of the following analgesic medications is considered the drug of choice for the treatment of prolonged moderate to severe pain for children with cancer? *(morphine)* | 12; 0.92 | 12; 0.92 | | Kept; – |
| Q29_ Which of the following IV doses of morphine administered would be equivalent to 15 mg of oral morphine? *(morphine 5mg IV)* | 11; 0.85 | 10; 0.77 | Q30_ Which of the following IV morphine doses is approximately equivalent to 15 mg of oral morphine? *(morphine 5mg IV)* | 11; 0.85 | 10; 0.77 | Requires knowledge of pharmacology (n = 1). | Kept; – |
| Q30_Analgesics for post-operative pain should initially be given: *(around the clock on a fixed schedule)* | 11; 0.85 | 13; 1.00 | Q31_Analgesics for post-operative pain should initially be given: *(around the clock on a fixed schedule)* | 11; 0.85 | 13; 1.00 | | Kept; – |
| Q31_A child with chronic cancer pain has been receiving daily opioid analgesics for 2 months. The doses increased during this time period. Yesterday the child was receiving morphine 20 mg/hour intravenously. Today he has been receiving 25 mg/hour intravenously for 3 hours. The likelihood of the child developing clinically significant respiratory depression is: *(<1%)* | 11; 0.85 | 11; 0.85 | | | | | |
| Q32_Analgesia for chronic cancer pain should be given: *(around the clock on a fixed schedule)* | 13; 1.00 | 13; 1.00 | Q32_ Analgesia for chronic cancer pain should be given: *(around the clock on a fixed schedule)* | 13; 1.00 | 13; 1.00 | | Kept; – |
| Q33_The most likely explanation for why a child/adolescent with pain would request increased doses of pain medication is: *(the child/adolescent is experiencing increased pain)* | 12; 0.92 | 12; 0.92 | Q33_The most likely reason a child/ adolescent with pain would request increased doses of pain medication is: *(the child/ adolescent is experiencing increased pain)* | 12; 0.92 | 12; 0.92 | Can use child alone (n = 1). | Kept; – |
| Q34_ Which of the following drugs are useful for treatment of cancer pain? *(all of the above)* | 12; 0.92 | 11; 0.85 | Q34_Which of the following drugs are potentially useful for treatment of children's cancer pain? *(all of the above)* | 12; 0.92 | 11; 0.85 | | Kept; – |
| Q35_The most accurate judge of the intensity of the child's/adolescent's pain is: *(the child/ adolescent)* | 13; 1.00 | 11; 0.85 | Q35_The most accurate judge of the intensity of the child's/ adolescent's pain is the: *(child/ adolescent)* | 13; 1.00 | 11; 0.85 | | Kept; – |

*(Continued)*

**Table 2.** (*Continued*)

| Items on the PNKAS 1999 version (*Answer*) | PNKAS | | Items on the PHPKASRP 2014 revised version (*Answer*) | PHPKASRP | | Pediatric Experts' Comments (Number of Experts) | Action Taken; Revised Form |
|---|---|---|---|---|---|---|---|
| | Relevance; I-CVIs | Clarity; I-CVIs | | Relevance; I-CVIs | Clarity; I-CVIs | | |
| Q36_Which of the following describes the best approach for cultural considerations in caring for a child/ adolescent in pain? a. Because of the diverse and mixed cultures in the United States, there are no longer cultural influences on the pain experience. b. Nurses should use knowledge that has defined clearly the influence of pain on culture (e.g., Asians are generally stoic, Hispanics are expressive and exaggerate their pain, etc.). c. *Children/ adolescents should be individually assessed to determine cultural influences on pain.* | 11; 0.85 | 12; 0.92 | Q36_Which of the following describes the best approach for cultural considerations in caring for a child/ adolescent in pain? a. There are no longer cultural influences on the pain experience in the United States due to the diversity of the population. b. Healthcare providers should use knowledge that has defined clearly the influence of pain on culture (e.g. Asians are generally stoic, Hispanics are expressive and exaggerate their pain, etc.) c. *Children/ adolescents should be individually assessed to determine cultural influences on pain.* | 11; 0.85 | 12; 0.92 | Examples given should be modified to suit the Ghanaian context or deleted to make it more generalized (n = 1). | **Amended;** Which of the following describes the best approach for cultural considerations in caring for child/ adolescent in pain? a. There are no longer cultural influences on the pain experience due to the diversity of the population. b. Nurses/ healthcare providers should use knowledge that has defined clearly the influence of pain on culture c. Children/ adolescents should be individually assessed to determine cultural influences on pain. |
| Q37_What do you think is the percentage of patients who over report the amount of pain they have? Circle the correct answer. (*0 or 10%*) | 8; 0.62 | 9; 0.69 | Q37_What do you think is the percentage of patients who over report the amount of pain they have? (*0 and 10%*) | 8; 0.62 | 9; 0.69 | On what basis are respondents expected to guess this percentage? (n = 1). | **Amended;** Children generally over report their pain. (*True/ **False***) |
| Q38_Narcotic/ opioid addiction is defined as psychological dependence accompanied by overwhelming concern with obtaining and using narcotics for psychic effect, not for medical reasons. It may occur with or without the physiological changes of tolerance to analgesia and physical dependence (withdrawal). Using this definition, how likely is it that opioid addiction will occur as a result of treating pain with opioid analgesics? Circle the number closest to what you consider the correct answer. (*<1%*) | 12; 0.92 | 9; 0.69 | Q25_Opioid/ narcotic addiction is defined as a chronic neurobiological disease, characterized by impaired control over drug use, compulsive use, continued use despite harm, and craving. It may occur with or without the physiological changes of tolerance to analgesia and physical dependence (withdrawal). Given this information, all children /adolescents whose pain have been treated with opioids for longer than a month are addicted to opioids. (*False*) | 12; 0.92 | 9; 0.69 | Question should be straight forward (n = 1). | Kept; – |
| Q39_On the patient's record you must mark his pain on the scale below. Circle the number that represents your assessment of Andrew's pain. (*8*) | 12; 0.92 | 10; 0.77 | Q38_On the patient's record you must mark his pain on the scale below. Choose the number that represents your assessment of Andrew's pain. (*8*) | 12; 0.92 | 10; 0.77 | | Kept; – |

(*Continued*)

**Table 2.** (Continued)

| Items on the PNKAS 1999 version (*Answer*) | PNKAS | | Items on the PHPKASRP 2014 revised version (*Answer*) | PHPKASRP | | Pediatric Experts' Comments (Number of Experts) | Action Taken; Revised Form |
|---|---|---|---|---|---|---|---|
| | Relevance; I-CVIs | Clarity; I-CVIs | | Relevance; I-CVIs | Clarity; I-CVIs | | |
| Q40_Your assessment, above, is made two hours after he received morphine 2 mg IV. After he received the morphine, his pain ratings every half-hour ranged from 6 to 8 and he had no clinically significant respiratory depression, sedation, or other untoward side effects. He has identified 2 as an acceptable level of pain relief. His physician's order for analgesia is "morphine IV 1–3 mg q1h PRN pain relief." Check the action you will take at this time. (*administer morphine 3 mg IV now*) | 12; 0.92 | 13; 1.00 | Q39_Your assessment, above, is made two hours after he received morphine 2 mg IV. After he received the morphine, his pain ratings every half-hour ranged from 6 to 8 and he had no clinically significant respiratory depression, sedation, or other untoward side effects. He has identified 2 as an acceptable level of pain relief. His physician's order for analgesia is "morphine IV 1–3 mg q1h PRN pain relief." Check the action you will take at this time. (*administer morphine 3 mg IV now*) | 12; 0.92 | 13; 1.00 | | Kept; – |
| Q41_On the patient's record you must mark his pain on the scale below. Circle the number that represents your assessment of Robert's pain: (*8*) | 12; 0.92 | 11; 0.85 | Q40_Select the number that represents your assessment of Robert's pain: (*8*) | 12; 0.92 | 11; 0.85 | | Kept; – |
| Q42_Your assessment, above, is made two hours after he received morphine 2 mg IV. After he received the morphine, his pain ratings every half-hour ranged from 6 to 8 and he had no clinically significant respiratory depression, sedation, or other untoward side effects. He has identified 2 as an acceptable level of pain relief. His physician's order for analgesia is "morphine IV 1–3 mg q1h PRN pain relief." Check the action you will take at this time: (*administer morphine 3 mg IV now*) | 12; 0.92 | 11; 0.85 | Q41_Your assessment, above, is made two hours after he received morphine 2 mg IV. After he received the morphine, his pain ratings every half-hour ranged from 6 to 8 and he had no clinically significant respiratory depression, sedation, or other untoward side effects. He has identified 2 as an acceptable level of pain relief. His order for analgesia is "morphine IV 1–3 mg q1h PRN pain relief." Check the action you will take at this time: (*administer morphine 3 mg IV now*) | 12; 0.92 | 11; 0.85 | | Kept; – |

**NB:** PNKAS–Pediatric Nurses' Knowledge and Attitudes Survey Regarding Pain, PHPKASRP–Pediatric Healthcare Providers' Knowledge and Attitudes Survey Regarding Pain, PO–Per os (by mouth), mg–Milligram, IV–Intravenous, q1h –Hourly, PRN–When necessary.

scale (S-CVI/Ave) was 0.89, indicating an acceptable level of content validity which is almost at the 0.90 acceptable excellent value.

## Content validity assessment of the PHPKASRP instrument

The number of items considered relevant (with a rating of 3 or 4) by all 13 pediatric experts was five. The individual-item content validity index (I-CVI) for the 41 items ranged from 0.62 to 1.00 for the relevance component of the PHPKASRP instrument. Five (5) out of the 41

items fell below the recommended I-CVI of 0.70; thus, the proportion of items considered relevant on this basis was 87.8%. The average CVI relevance for the scale (S-CVI/Ave) was 0.86, indicating an acceptable level of content validity and slightly below the 0.90 accepted excellent value.

The number of items considered clear (with a rating of 3 or 4) by all 13 experts was 13. The individual-item content validity index (I-CVI) for the 41 items ranged from 0.62 to 1.00 for the clarity component of the PHPKASRP instrument. Three (3) out of the 41 items fell below the recommended I-CVI of 0.70; the resultant proportion of items considered clear on this basis was 92.7%. The average CVI clarity for the scale (S-CVI/Ave) was 0.89, indicating an acceptable level of content validity and almost at the 0.90 accepted excellent value.

## Comparison of the content validity of PNKAS and PHPKASRP instruments

The results revealed that both instruments have an acceptable level of content validity (refer to Table 3). Nevertheless, the PNKAS instrument performed slightly better than the PHPKASRP instrument in four areas. These were related to the following: the number of items considered relevant with an I-CVI $\geq$ 0.70 (90.5% versus 87.8%) and those considered clear with an I-CVI $\geq$ 0.70 (95.2% versus 92.7%); the average scale-level content validity index for the relevance component of the items (0.87 versus 0.86) and the range of items considered clear (0.69–1.00 versus 0.62–1.00).

## Experts' comments on the PNKAS and PHPKASRP instruments

The pediatric experts made general comments on both instruments (PNKAS and PHPKASRP) which included: simplifying the sentences, separating knowledge questions from those of attitude for clarity purposes, and restructuring questions according to the different pediatric pain topics. They additional made specific comments on the individual items which have been presented in Table 2. In consultation with the instrument developer (Manworren, R.C.B.), the researchers addressed all comments from the experts through a review process which resulted in the maintenance of 37 items and revision of 5 items on the PNKAS instrument and the maintenance of 36 items and revision of 5 items on the PHPKASRP instrument.

**Table 3. Comparison of the content validity of PNKAS and PHPKASRP instruments (n = 13).**

| Variable | PNKAS (42 items) | PHPKASRP (41 items) |
|---|---|---|
| *Relevance Components* | | |
| Universal Agreement | 5 | 5 |
| Number of Items with I-CVI $\geq$ 0.70 | 38 (90.5%) | 36 (87.8%) |
| Number of Items with I-CVI < 0.70 | 4 (9.5%) | 5 (12.2%) |
| Minimum–Maximum I-CVI | 0.62–1.00 | 0.62–1.00 |
| SCI/Ave | 0.87 | 0.86 |
| *Clarity Components* | | |
| Universal Agreement | 13 | 13 |
| Number of Items with I-CVI $\geq$ 0.70 | 40 (95.2%) | 38 (92.7%) |
| Number of Items with I-CVI 0.70 | 2 (4.8%) | 3 (7.3%) |
| Minimum–Maximum I-CVI | 0.69–1.00 | 0.62–1.00 |
| SCI/Ave | 0.89 | 0.89 |

**NB:** I-CVI–Individual item level content validity index, SCI/Ave–Average scale-level content validity index

## Discussion

The current study aimed at comparing the content validity of two versions of an instrument used in measuring pediatric pain knowledge and attitudes (PNKAS and PHPKASRP). Our results showed that both instruments have an acceptable level of content validity, signifying that the instruments sufficiently represent the content of "pediatric pain knowledge and attitudes" for which they intend to measure [32]. Nevertheless, the PNKAS instrument performed slightly better than the PHPKASRP instrument with regards to some aspects of its content validity properties. For instance, the number of items with I-CVI $\geq$ 0.70 for both relevance and clarity aspects of PNKAS was 90.5% and 95.2% as against 87.8% and 92.7% for PHPKASRP respectively. On the basis of the current study findings, preference may be given for the use of the PNKAS due to its slightly higher level of content validity and extensive content. However, both instruments are comparable in the length of time they take to be completed. Thus, both instruments are promising in being used in clinical practice and for research purposes due to their acceptable level of content validity.

A critical review of both instruments (PNKAS and PHPKASRP) seems to be missing some important aspects of pediatric pain management, especially on the role of family caregivers in children's pain management. The role of family caregivers has been shown to be critical in pediatric pain management as they serve as the mouthpiece for vulnerable children [38, 39], especially in the Ghanaian society where children are expected to be "seen" but not "heard" [40]. The pediatric experts also brought to the fore their unfamiliarities with given examples of nonpharmacological pain management interventions (such as guided imagery, biofeedback among others) and the need to contextualize pediatric pain care considerations to the Ghanaian setting instead of referring to other countries such as United States and foreign ethnic origins such as Hispanics, Asians and so on. This underscores the importance of validity assessment as there exists universal differences in culture, semantics and resources in different parts of the world [19, 20]. It further supports the assessment of content validity as a precondition for other forms of validity such as construct and criterion validity [41].

The production of high-quality data in quantitative research requires thorough evaluation of an instrument to build sufficient evidence for its validity [27]. Content validity testing is thus, concerned with determining the inferences that can be made about an instrument's construction. The processes involved in the content validity assessment has led to an improvement in clarity and relevance of the items contained in these instruments. This process has also provided empirical data supporting the adaptation process [42], which will also facilitate subsequent testing of the instrument for other types of validity and reliability [43]. In line with the recommendations of Delgado-Rico and colleagues [34], decisions on items (i.e., elimination, modification or conservation) were made on the basis of the content validity indices, feedback given by experts, inputs from the instrument developer and the contribution of the items to the overall construct under investigation. At the end of the validation processes, 37 and 36 items were respectively kept on the PNKAS and PHPKASRP instruments whereas 5 items were modified on each of the two instruments.

Feasibility of the instruments was reflected in the high response rate and the absence of missing values as all the experts completely filled the data collection instruments. On the basis of the current study findings, we recommend the use of the either of these instruments (PNKAS or PHPKASRP) as one that sufficiently covers the construct of pediatric pain knowledge and attitudes. Nonetheless, preference may be given to the PNKAS instrument for use in clinical practice and research due to its slightly higher level of content validity. On the basis of the current study findings, we recommend the use of either one of two instruments (PNKAS

and PHPKASRP) in the proposed pediatric pain education for nursing students and nurses in Ghana due to their acceptable level of content validity.

One of the short-comings of the current study was our inability to conclude on the validity of these revised instruments as they were not tested; future studies should examine the content validity of these revised instruments. Other forms of validity (construct and criterion) and reliability (internal consistency, test-retest, intrarater) should also be evaluated in the future to enhance our understanding on their psychometric properties. It is also worth mentioning that our method of content validity assessment did not cater for the possibility of chance agreement among the experts which is achieved using Kappa statistic coefficient [32, 44].

## Conclusions

The PNKAS and PHPKASRP instruments have demonstrated an acceptable level of content validity in the Ghanaian context. Both instruments sufficiently cover the construct of "pediatric pain knowledge and attitudes". We recommend the use of either of these two instruments for use in clinical practice and research purposes. The modifications made on these two instruments should be assessed for content validity in the future. Other forms of validity (construct, criterion) and reliability (internal consistency, test-retest, item analysis) should also be examined in future studies.

## Supporting information

**S1 Appendix. Revised Pediatric Nurses' Knowledge and Attitudes Survey regarding pain (r-PNKAS).**
(DOCX)

**S2 Appendix. Revised Pediatric Healthcare Providers' Knowledge and Attitudes Survey Regarding Pain (r-PHPKASRP).**
(DOCX)

**S1 Data.**
(XLSX)

## Acknowledgments

We are grateful to the pediatric experts who took time off their very busy schedules to participate in the study. Many thanks to Renee C. B. Manworren for her valuable inputs in the instrument modification process.

## Author Contributions

**Conceptualization:** Abigail Kusi Amponsah, Victoria Bam, Anna Axelin.

**Data curation:** Abigail Kusi Amponsah.

**Formal analysis:** Abigail Kusi Amponsah, Victoria Bam, Minna Stolt, Joonas Korhonen, Anna Axelin.

**Funding acquisition:** Abigail Kusi Amponsah, Victoria Bam, Minna Stolt, Joonas Korhonen, Anna Axelin.

**Investigation:** Abigail Kusi Amponsah, Victoria Bam, Minna Stolt, Joonas Korhonen, Anna Axelin.

**Methodology:** Abigail Kusi Amponsah, Victoria Bam, Minna Stolt, Joonas Korhonen, Anna Axelin.

**Project administration:** Abigail Kusi Amponsah, Victoria Bam, Anna Axelin.

**Resources:** Abigail Kusi Amponsah, Victoria Bam, Minna Stolt, Joonas Korhonen, Anna Axelin.

**Software:** Abigail Kusi Amponsah.

**Supervision:** Victoria Bam, Anna Axelin.

**Validation:** Abigail Kusi Amponsah, Victoria Bam, Minna Stolt, Joonas Korhonen, Anna Axelin.

**Visualization:** Abigail Kusi Amponsah, Victoria Bam, Minna Stolt, Joonas Korhonen, Anna Axelin.

**Writing – original draft:** Abigail Kusi Amponsah.

**Writing – review & editing:** Abigail Kusi Amponsah, Victoria Bam, Minna Stolt, Joonas Korhonen, Anna Axelin.

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
