## [Decision Letter · Decision Letter 0]

22 Apr 2020

PONE-D-20-05637

Comparison of the content validity of two instruments used in measuring pediatric pain knowledge and attitudes

PLOS ONE

Dear Ms. Kusi Amponsah,

Thank you for submitting your manuscript to PLOS ONE. After careful consideration, we feel that it has merit but does not fully meet PLOS ONE’s publication criteria as it currently stands. Therefore, we invite you to submit a revised version of the manuscript that addresses the points raised during the review process.

ACADEMIC EDITOR: 

Editor Decision - Major Revision

Please, follow all reviewers commentaries.

We would appreciate receiving your revised manuscript by Jun 06 2020 11:59PM. To enhance the reproducibility of your results, we recommend that if applicable you deposit your laboratory protocols in protocols.io, where a protocol can be assigned its own identifier (DOI) such that it can be cited independently in the future. For instructions see: http://journals.plos.org/plosone/s/submission-guidelines#loc-laboratory-protocols

We look forward to receiving your revised manuscript.

Kind regards,

Matias Noll, Ph.D

Academic Editor

PLOS ONE

Journal Requirements:

Additional Editor Comments (if provided):

Reviewers' comments:

Reviewer's Responses to Questions

**Comments to the Author**

1. Is the manuscript technically sound, and do the data support the conclusions?

Reviewer #1: Yes

Reviewer #2: Yes

Reviewer #3: Partly

2. Has the statistical analysis been performed appropriately and rigorously? 

Reviewer #1: Yes

Reviewer #2: I Don't Know

Reviewer #3: Yes

3. Have the authors made all data underlying the findings in their manuscript fully available?

Reviewer #1: No

Reviewer #2: Yes

Reviewer #3: Yes

4. Is the manuscript presented in an intelligible fashion and written in standard English?

Reviewer #1: Yes

Reviewer #2: Yes

Reviewer #3: No

5. Review Comments to the Author

Reviewer #1: 1 Summary of the research and your overall impression

1.1 Reviewer comment:

The manuscript compared the content validity of two instruments used in measuring nurse pediatric pain knowledge and attitudes.

As strong points of the article, it should be noted that they have had a wide participation of experts. Furthermore, it appears that the author of the questionnaire, Manworren, has been involved in the counseling process.

As weaknesses, the authors acknowledge both in the limitations section and in the conclusions that, as future lines, a validation of the criterion and the construct is required.

From my point of view, we must differentiate between (Lobiondo-Wood G, Haber J.; 2013) :

• The content validity. Content validity evaluates qualitatively whether the questionnaire covers all the dimensions of the phenomenon that wants to measure, since it is considered that an instrument is valid in its content if contemplates all related aspects with the concept that measures. The researcher begins by defining the concept and identifying the attributes or dimensions of the concept. The items that reflect the concept and its domain are developed.

• The apparent validity. It is a subtype of content validity. It is a face validity, which is a rudimentary type of validity that basically verifies that the instrument fives the appearance of measuring the concept. It is an intuitive type of validity in which colleagues or subjects are asked to read the instrument and evaluate the content in terms of whether it appear to reflect the concept the researcher intends to measure. Or if the elements included in an instrument are relevant.

• Criterion validity. It is the degree of correlation between an instrument and another measure of the variable under study that serves as criterion or reference.

• Construct validity. It is understood as the degree to which an instrument measures the bipolar evaluative dimension for which it was designed.

Actually, in the present article, what they do is to measure apparent validity. It is important because the acceptance of a scale by several people gives consistency when using it. However, apparent and content validity is a relevant method especially when designing an instrument. It is not so important when the instrument has been previously validated and used in different areas.

On the other hand, and as the authors say, these tools have already been validated and used in various studies. Knowing that its content validity has already been reviewed as they describe and explain. In that case, why have they revised the apparent validity again? Why haven't they gone a step further? Why have the criteria not been applied and reviewed?

For these reasons, I think the article does not reach the level required for a journal like PLOS ONE, and should be rejected.

From my point of view, the approach and content is well developed, but due to the characteristics of the tools used, these require a criterion validity, a construct validity, or a cross-cultural validation.

The authors summarize the main research question and key findings. Even, the authors identify other literature on the topic and explain how the study relates to this previously published research.

However, I would like to make some specific suggestions in the next point.

2 Discussion of specific areas for improvement

2.1 Major issues

2.1.1 Reviewer comment:

Suggestions for improvement do not refer to major issues

2.2 Minor issues

2.2.1 Reviewer comment:

2.2.1.1 Title

As a suggestion, and to contextualize more the article from the first moment, the title could refer to Pediatric Healthcare Providers' (HCPs) or pediatric nurses as well as keywords.

In fact, HCPs are referenced every time in the page 3, line 61 paragraph.

Another more clarifying explanation is found in page 3 and line 68: “These two instruments were developed to measure healthcare professionals and students’ knowledge and attitudes regarding children’s pain [13].” But it is still a personal appreciation.

2.2.1.2 Abstract and introduction

Page 2; line 24. The introduction does not set the stage adequately. As in the title, it is required to specify who the study is aimed at. To miss this information may decontextualize and imply the study population.

In the abstract, the authors explain why the study matters and put the research in context properly. However:

In page 2; line 25; the authors clarify that “This was considered necessary due to the universal differences in culture, semantics and healthcare resources in different parts of the globe”, but in the following paragraph:

In page 2; line 31; they specify that the experts only will check the relevance and clarity of the items will be reviewed without mention culture, semantics and healthcare resources. If it is so important, because you mention it at the beginning of the abstract, could the experts have been asked about the changes due to the cultural factor? Has any change been made due to semantic and cultural changes?

Content validity is a relevant method especially when designing an instrument. It is not so important when the instrument has been previously validated and used in different areas. However, when an instrument is translated into another language, if the explored concepts are supposed to change significantly from one culture to another, it may be useful to recheck the face validity.

Moreover, in page 4, line 92 It is said that validity is not the property of an instrument, but depends on the interpretation related to the context and participants. However, knowledge of health is based on science, evidence, principles, theories, and is universal. Therefore, they do not depend on the interpretation of people and cultures.

Content validity evaluates qualitatively whether the questionnaire covers all the dimensions of the phenomenon to be measured, since an instrument is considered to be valid in its content if it considers all aspects related to the concept it measures. For this, it is necessary to have a clear idea of the conceptual aspects to be measured. And in this case, since the instrument had previously been validated and used in different areas, the content was already available.

Page 3; line 67. This is what I cannot understand. In this section, the questions that arise are: When talking about the revised version (PHPKASRP). Is the article review being done for the first time in this article? Or has this version already been created before? Why do you call short version if they have almost the same number of questions?...May be could you give more details or change the way to explain it.

In the next sentence: page 3; line 68. “These two instruments were developed to measure healthcare professionals and students’ knowledge and attitudes regarding children’s pain (13)”. We find the reference of the PNKAS, what about the PHPKASRP reference? Could you explain this better?

When talking about instrument validation, I would give more information about whether or not PHPKASRP is validated, etc.

Page 4; line 84. References for PHPKASRP are again needed.

2.2.1.3 Figures and tables

The information in Tables 1 and 3 are explained in the text. Table 1 could be omitted by completing the information in the text. However, Table 3 can be kept, since it serves as a synthesis of the results.

2.2.1.4 Methods

Page 6; paragraph line 128-133.There is no specific reference to cultural and semantic factors. Despite the emphasis that has been given in the introduction by providing references.

They talk about: comprehensiveness, objectivity, organization and relevance defined as “comprehensive: that issue containing important information to reach the objective of the study, stated in a comprehensible manner; Objective: that issue which is easy to understand; organization: the disposition of the issues and alternatives as well as their content; Relevant: that question which is related to achieving the goal of the research”

Page 8; line 182. Again, the article reference of the revised version (PHPKASRP instrument) is missed.

2.2.1.5 Results, discussion, conclusions

The results on page 10 should reference Table 3, where the results are summarized.

The authors acknowledge both the limitations and the conclusions that, as future lines, a validation of the criterion and the construct is required.

2.2.1.6 Statistical analysis

This type of validity requires basic statistics. Experts and / or researchers use in evaluating the relevance of a scale that requires descriptive analysis.

Reviewer #2: This descriptive methodological study was carried out with the objective of evaluating and comparing the content validity of two instruments of knowledge and attitude about pediatric pain.

The study was well written and relevant, however, my main concerns are related to Validity analysis. The lack of review of the validity of the content of these instruments and the lack of reliability analysis (internal consistency, test-retest, intra-evaluator). In addition, the analysis of casual agreement among experts, with the Kappa statistical coefficient.

Type of study: Review the written form of the type of study. "Descriptive methodological study with content validity".

Review confusing writing in the first paragraph of the results.

Reviewer #3: Dear Editor

Thank you for the opportunity to review this manuscript. The manuscript aims to evaluate and compare the content validity of the PNKAS and its revised version, the PHPKASRP from a Ghanaian perspective. I have read the manuscript with careful attention. I have several major and minor comments for better improvement. The authors are expected to make point-by-point response to the comments.

6. PLOS authors have the option to publish the peer review history of their article (what does this mean?). If published, this will include your full peer review and any attached files.

Reviewer #2: No

Reviewer #3: No

---

## [Author Response · Author response to Decision Letter 0]

6 Jun 2020

Response to Review Comments

Thank you for taking time off your busy schedules and providing us constructive feedback to further improve our manuscript. Please find below our responses to the review comments. 

Overall Comments from Reviewer 1

This descriptive methodological study was carried out with the objective of evaluating and comparing the content validity of two instruments of knowledge and attitude about pediatric pain.

The study was well written and relevant, however, my main concerns are related to Validity analysis. The lack of review of the validity of the content of these instruments and the lack of reliability analysis (internal consistency, test-retest, intra-evaluator). In addition, the analysis of casual agreement among experts, with the Kappa statistical coefficient.

Response

Thank you for the feedback. The review of validity and reliability of the instruments have already been provided on page 4, lines 75-87.

Specific Comments from Reviewer 1

Comment

Type of study: Review the written form of the type of study. "Descriptive methodological study with content validity".

Response

The study type has been amended on page 5, line 103.

Comment

Review confusing writing in the first paragraph of the results.

Response

Thank you for the feedback. The review of validity and reliability of the instruments have already been provided on page 4, lines 75-87.

Overall Comments from Reviewer 2

The manuscript aims to evaluate and compare the content validity of PNKAS and its revised version, PHPKASRP from a Ghanaian perspective. I have read the manuscript with careful attention. First of all, I was expecting to find the revised the PNKAS and PHPKASRP) as appendices. I think it is worthy to include them as appendices since they are the final products of the study. I have other comments for better improvement.

Response

Thank you for the feedback. The revised instruments have been provided as appendices (refer to S1 Appendix and S2 Appendix).

Specific Comments from Reviewer 1

Comment

The title of the manuscript reads as if the study compares completely different instruments. However, PHPKASRP is a revised version of PNKAS, and it would be not surprising to find high similarity between the two versions in terms of content validity. Therefore, it would be irrational making a comparison between two versions of the same survey. Further, the title does not indicate that the assessment of content validity was performed in Ghanaian context.

I would suggest revising the title to address these two comments. For example “Evaluating the content validity of two instruments used in measuring pediatric pain knowledge and attitudes in Ghanaian context”.

Response

As suggested, changes have been made to the titles (long and short) as found on page 1, lines 1-7.

Comment

Since the keyword “psychometric evaluation” might indicates reliability, validity, and/or responsiveness, the authors only examined the content validity. I believe this keyword would be misleading. Please replace it with “content validity”. Furthermore, I would suggest including “Ghana” as a keyword.

Response

Changes have been made on page 3, line 54.

Comment

Line 24-25. “we compared the content validity of two instruments used in measuring pediatric pain knowledge and attitudes”

Again, this sentence reads as if the authors compared completely different instruments, and it does not indicate that the evaluation was conducted in Ghanaian context. I would suggest recast this sentence as “we evaluated the content validity of two versions of an instrument used in measuring pediatric pain knowledge and attitudes in the Ghanaian context”.

Response

Changes have been made at the abstract and sections of the manuscript. Refer to page 2, lines 24-25, 28-29 and page 5, line 106.

Comment

Line 28-30. “Thirteen (13) pediatric experts in Ghana assessed the content validity of two instruments: the 42-item Pediatric Nurses' Knowledge and Attitudes Survey Regarding Pain (PNKAS) and the 41-item Pediatric Healthcare Providers’ Knowledge and Attitudes Survey Regarding Pain (PHPKASRP).”

Same here. I suggest recast this sentence as “Thirteen (13) pediatric experts in Ghana assessed the content validity of the 42-item Pediatric Nurses' Knowledge and Attitudes Survey Regarding Pain (PNKAS) and its revised version, the 41-item Pediatric Healthcare Providers’ Knowledge and Attitudes Survey Regarding Pain (PHPKASRP).

If you accept this suggestion, please modify the rest of abstract and manuscript to be consistent with these changes.

Response

Changes have been made at the abstract and sections of the manuscript. Refer to page 2, lines 24-25, 28-29 and page 5, line 106.

Comment

Line 31. “four-point likert scaled”. Please capitalize the letter “L” in “Likert”.

Response

As suggested, this has been changed on page 2, line 32.

Comment

Line 66-68. “Key among them are Pediatric Nurses’ Knowledge and Attitudes Survey regarding pain (PNKAS) and its revised shorter version, the Pediatric Healthcare Providers’ Knowledge and Attitudes Survey Regarding Pain (PHPKASRP)”

I know the PNKAS was published by Manworren RC (reference 13). However, was the PHPKASRP also published in the same study (reference 13)? Either way, you need to cite the study/studies that published the PNKAS and PHPKASRP in the above sentence.

Further, since the difference between the two versions in terms of number of items is one item only (42 vs 41), I am not sure if it is appropriate to describe the PHPKASRP as “revised shorter version”. I suggest deleting “shorter” form the above sentence.

Response Citations have been respectively placed by each instrument, refer to page 4, lines 71-73. Information on the PHPKASRP was provided by the instrument developer (Manworren RCB) through email communication. According to the instrument developer, she is yet to publish information about PHPKASRP.

Also, the word “shorter” has been removed from the sentence, refer to page 4, line 71.

Comment

Line 68-70. “These two instruments were developed to measure healthcare professionals and students’ knowledge and attitudes regarding children’s pain [13]”

This sentence reads as if Manworren RC (reference 13) published the two instruments in their study. However, I am not sure if this is true. Please make sure and correct the reference accordingly.

Response

Modifications have been made on page 3, line 70 and page 4, lines 71-74.

Comment

Line 72. “Following the development of the PNKAS instrument”

No need to follow the PNKAS or the PHPKASRP with the word “instrument” since the last word both terms is “survey”. Please delete “instrument” from this sentence and throughout the manuscript if you accept.

Response

All instances of the word “instrument(s)” which followed the PNKAS and or PHPKASRP have been removed from the manuscript.

Comment

Line 73-78. “Face and content validity of the instrument was established by five pain management experts in the United States of America. Internal consistency of the PNKAS instrument has also been assessed by two distinct groups with a reported acceptable Cronbach’s alpha value of 0.72 among 247 pediatric nurses and 0.77 among 88 members of a children’s nursing organization. Test-retest reliability analysis among 12 clinicians (6 nurses and 6 child life specialists) recorded a correlation coefficient of 0.67, signifying an acceptable level of instrument stability”.

Too much information are mentioned in this section. You may simply report the psychometric properties without going into further details about the number of participants or experts and their genders.

Response

As recommended, this statement has been summarised concisely; refer to page 4, lines 75-79.

Comment

Line 85. “(PHPKASRP instrument)”

Same here. Please see my comment above about the word “instrument”.

Response

As earlier indicated, all instances of the word “instrument(s)” which followed the PNKAS and or PHPKASRP have been removed from the manuscript.

Comment

Line 85-87. “Content validity of the PHPKASRP has been established by national content experts comprising of physicians, pediatric nurses and pharmacists in the United States of America”. Please cite the reference for this sentence.

Response

Citation to this statement has been provided on page 4, line 87.

Comment

Line 87-89. “According to the instrument developer (Manworren, R.C.B.), the instrument has been translated into other languages and permitted for use in many organizations around the world.”. Same here. Please cite the reference for this sentence.

Response

Citation to this statement has been provided on page 4, line 89.

Comment

Line 96-98. “As part of plans to use a pediatric pain knowledge and attitude instrument as a tool in assessing the effectiveness of a short-course educational program for nursing students and nurses”

According to this sentence, the content validity of the two versions was examined for the purpose of this course. Will be the course conducted in Ghana? Are all of the audience from Ghana? Or others from different countries will be included?

Response

Clarifications on this have been provided at page 5, line 98.

Comment

Line 98. “the current study sought to evaluate and compare the content validity”. Again, comparing the content validity of two versions of the same instrument would be irrational. I suggest narrowing your aim to “evaluating” without “comparing”.

Response

Suggested amendment has been made on page 5, line 98.

Comment

Line 103. “This descriptive methodological study”

According to Portney and Watkins (Foundations of Clinical Research: Applications to Practice. Vol. 2. Prentice Hall Upper Saddle River, NJ; 2000), validity studies are described as exploratory methodological studies. Please correct the design.

Response

The study type has been amended on page 5, line 103.

Comment

Line 110-111. “The experts were chosen based on their level of training, clinical and/ or teaching experience in pediatrics”.

The criteria of selecting the experts mentioned here is wide and lacks of accuracy. What do you mean by “their level”? Did you have minimum level of experience? Please be more specific.

I think it is very important to be more careful in describing how the experts were selected because they are the main element of this study.

Response

Specifications about the expertise have been provided on page 5, lines 111-112.

Comment

Line 135. “Both individual item-level and scale level content validity indices”

There are several issues in the manuscript regarding the use of abbreviations as will be stated in some of the following comments. As a rule of thumb, you should explain each of your abbreviation the first time it appears in the main text, and then use that abbreviation instead of the complete term in the rest of the manuscript.

In this sentence, you mentioned the individual item-level content validity (I-CVI) and the average scale level content validity (S-CVI/Ave) for the first time. Therefore, you should have introduced their abbreviations and use them instead of the complete term in the rest of your manuscript.

Line 149. “Chicago, IL”

The correct citation of the SPSS version 25 is (Armonk, NY). Please refer to this webpage for more information (https://www.ibm.com/support/pages/how-cite-ibm-spss-statistics-or-earlier-versions-spss).

Response

As suggested, amendments have been made throughout the manuscript; refer to page 6, line 134 for an example. 

The appropriate citation for SPSS version 25 have been provided on page 7, line 147.

Comment

Line 182-184. “The Pediatric Nurses' Knowledge and Attitudes Survey Regarding Pain (PNKAS) is a modification of Nurses' Knowledge and Attitudes Survey Regarding Pain (NKASRP)”.

You do not need to use the complete terms since you have already introduced their abbreviations in the introduction (line 67-68).

Response

Suggested amendment has been made at page 7, line 156.

Comment

Line 191-193. “Ethical approval for the study with reference number CHRPE/AP/574/18 was provided by the Committee on Human Research, Publications and Ethics (CHRPE), School of Medical Sciences (SMS), Kwame Nkrumah University of Science and Technology (KNUST)”.

One the other hand, you do not need to state the abbreviations of the Committee on Human Research, Publications and Ethics, School of Medical Sciences, and Kwame Nkrumah University of Science and Technology since they were only mentioned once in the manuscript.

Response

As suggested, these abbreviations have been removed; refer to page 8, lines 189-190.

Comment

Line 205. “Majority of them were female (69%)”. Please correct the percentage to (69.2%) to be consistence with Table 1.

Response

Suggested modification has been provided on page 9, line 201.

Comment

Table 1. Is there is any reason why you used the median instead of the mean to report the central tendency?

Response

The rationale for the use of median has been provided on page 7, lines 147-149.

Comment

Line 215-216. “The average relevance CVI for the scale (S-CVI/Ave) was 0.87”.

Same to my previous comments about the abbreviations. Use the abbreviation without introducing it again. Please correct accordingly.

Response

As suggested, changes have been made throughout the manuscript. An instance of this can be seen on page 9, lines 209, 214 among others.

Comment

Line 232. “individual-item content validity index (I-CVI)”. Same here regarding the use of complete term.

Response

Modifications have been provided throughout the manuscript. Instances of this can be seen on page 9, line 219 and page 10, line 223.

Comment

Table 2. Some issues on this table need to be addressed:

The table is lengthy and I think it would be better to include it as a supplemental table instead of containing it in the manuscript.

There are four cells that are missing in the last column (Action Taken; Revised Form) of the table.

You mentioned five items were removed from each of the PNKAS and PHPKASRP. It is not clear from the table which items are they. Please specify them.

It is not clear why some items were retained or not revised although their scores were less than 70 and some of them received comments from the experts. For example, the question about respiratory depression and the use of opioids, Questions 18, the question about Anxiolytics, sedatives and barbiturates, Question 24 of the PHPKASRP, and Question 38 of the PNKAS (Q25 on the PHPKASRP). I think the readers deserve to know the reasons behind ignoring the low scores and experts’ comments in making the decision of retaining or not revising those items. Otherwise, the ratings and comments of the experts are meaningless.

Response

As suggested, Table 2 has been designated as S1 Table in the supplementary document file named “S1_Table.docx”. 

Missing information has been provided at the initial four empty cells of the now “S1 Table”.

Five items were modified from each of the instruments and “not removed”; these items have been specified on page 11, lines 249-251. 

Factors which influenced the decisions made on items have been provided on page 6, lines 138-140. 

Comment

Line 254-256. “In consultation with the instrument developer (Manworren, R.C.B.), the researchers addressed all comments from the experts through a review process”

This a bit confusing. In the methods section, line 139-142, the authors stated, “items which were < 0.70 were eliminated, revised or kept based on feedback and its importance to the construct under investigation. Items which had an I-CVI value of ≥ 0.70 were kept or modified based on theoretical relevance and/ or the feedback received from the participating pediatric experts”. Nothing were mentioned in the methods about involving the developer in the decision-making process.

Response

This has been rectified on page 6, lines 138-140.

Comment

Table 3. What do you mean by “universal agreement”? How this outcome was calculated?

Response

Definition and calculation of “universal agreement” has been provided on page 6, lines 141-142.

Comment

Line 311-314. “In line with the recommendations of Delgado-Rico and colleagues [43], decisions on items (i.e., elimination, modification or conservation) were made on the basis of the content validity indices, feedback given by experts, inputs from the instrument developer and the contribution of the items to the overall construct under investigation”.

Here you mentioned the involvement of the developer in the decision-making process. Again, nothing was mentioned about the developer involvement in the methods. Further, I think you need to cite Delgado-Rico and colleagues as well in your methods since you followed their recommendations in your decisions making.

Response

As earlier indicated, these concerns have been addressed, refer to page 6, lines 138-140.

Comment

Line 318-319. “we recommend the use of the both instruments (PNKAS and PHPKASRP) as simple, easy to use”

There is no evidence of simplicity nor easiness of use that can be drawn from you results.

Response

These descriptions have been removed; refer to page 12, lines 288-289.

Comment

Line 355. “Both instruments are simple, easy to use”

Same here, you need to show the evidence or delete this sentence.

Response

Suggested modifications have been made on page 13, lines 304-305.

Comment

I do not believe it is required to include the publishers in the reference (e.g., Lippincott, Williams & Wilkins, Elsevier, Hindawi ..etc.). The authors need to format their references according to the journal’s instructions (https://journals.plos.org/plosone/s/submission-guidelines#loc-references).

Response

All instances of the “publishers” have been removed to comply with the journal requirements; refer to pages 14-18.

Comment

There are several places throughout the manuscript with extra spaces between the words (e.g., line 39, line 41). Please correct the text.

Response

Extra spaces have been removed from the entire manuscript.

Overall Comments from Reviewer 3

The manuscript compared the content validity of two instruments used in measuring nurse pediatric pain knowledge and attitudes.

As strong points of the article, it should be noted that they have had a wide participation of experts. Furthermore, it appears that the author of the questionnaire, Manworren, has been involved in the counseling process.

As weaknesses, the authors acknowledge both in the limitations section and in the conclusions that, as future lines, a validation of the criterion and the construct is required.

From my point of view, we must differentiate between (Lobiondo-Wood G, Haber J.; 2013) :

• The content validity. Content validity evaluates qualitatively whether the questionnaire covers all the dimensions of the phenomenon that wants to measure, since it is considered that an instrument is valid in its content if contemplates all related aspects with the concept that measures. The researcher begins by defining the concept and identifying the attributes or dimensions of the concept. The items that reflect the concept and its domain are developed.

• The apparent validity. It is a subtype of content validity. It is a face validity, which is a rudimentary type of validity that basically verifies that the instrument fives the appearance of measuring the concept. It is an intuitive type of validity in which colleagues or subjects are asked to read the instrument and evaluate the content in terms of whether it appear to reflect the concept the researcher intends to measure. Or if the elements included in an instrument are relevant.

• Criterion validity. It is the degree of correlation between an instrument and another measure of the variable under study that serves as criterion or reference.

• Construct validity. It is understood as the degree to which an instrument measures the bipolar evaluative dimension for which it was designed.

Actually, in the present article, what they do is to measure apparent validity. It is important because the acceptance of a scale by several people gives consistency when using it. However, apparent and content validity is a relevant method especially when designing an instrument. It is not so important when the instrument has been previously validated and used in different areas.

On the other hand, and as the authors say, these tools have already been validated and used in various studies. Knowing that its content validity has already been reviewed as they describe and explain. In that case, why have they revised the apparent validity again? Why haven't they gone a step further? Why have the criteria not been applied and reviewed?

For these reasons, I think the article does not reach the level required for a journal like PLOS ONE, and should be rejected.

From my point of view, the approach and content is well developed, but due to the characteristics of the tools used, these require a criterion validity, a construct validity, or a cross-cultural validation.

The authors summarize the main research question and key findings. Even, the authors identify other literature on the topic and explain how the study relates to this previously published research.

However, I would like to make some specific suggestions in the next point.

Response

Thanks for the feedback. As you already know, a valid instrument is a reliable one but a reliable instrument does not guarantee validity. Thus, validity is an important quality expected of instruments. As you rightly point out and explain, there are several types of validity (content, construct, criterion, discriminant among others). The choice on the type of validity to be assessed depends on the aim of this enterprise. Our reasons for assessing content validity as the starting point for other types of validity and reliability in the future have been provided in the manuscript (on page 2, lines 25-27; page 4, lines 92-95 and page 5, lines 96-99). It is significant to also mention that, the changes which have been made in the instruments would have been missed if we had not assessed their content validity as the starting point for other types of validity and reliability assessments to be explored in the future.

Specific Comments from Reviewer 3

Comment

As a suggestion, and to contextualize more the article from the first moment, the title could refer to Pediatric Healthcare Providers' (HCPs) or pediatric nurses as well as keywords.

In fact, HCPs are referenced every time in the page 3, line 61 paragraph.

Another more clarifying explanation is found in page 3 and line 68: “These two instruments were developed to measure healthcare professionals and students’ knowledge and attitudes regarding children’s pain [13].” But it is still a personal appreciation.

Response

The titles (long and short) and keywords have been amended; refer to page 1, lines 1-7 and page 3, line 54.

Comment

Page 2; line 24. The introduction does not set the stage adequately. As in the title, it is required to specify who the study is aimed at. To miss this information may decontextualize and imply the study population.

In the abstract, the authors explain why the study matters and put the research in context properly. However:

In page 2; line 25; the authors clarify that “This was considered necessary due to the universal differences in culture, semantics and healthcare resources in different parts of the globe”, but in the following paragraph:

In page 2; line 31; they specify that the experts only will check the relevance and clarity of the items will be reviewed without mention culture, semantics and healthcare resources. If it is so important, because you mention it at the beginning of the abstract, could the experts have been asked about the changes due to the cultural factor? Has any change been made due to semantic and cultural changes?

Response

Thanks for the feedback. We however disagree with this comment as the introduction provides the background information and justification for the current study. We appreciate your suggestion to include the target population as part of the title but this would make it too lengthy as the PNKAS was specifically designed for nurses whilst the PHPKASRP was directed at pediatric healthcare professionals in general. 

Changes have been made to the assessable areas to reflect what happened in practice; this has been provided on page 6, lines 130-133.

At the end of the validation process, five items each were modified on the PNKAS and PHPKASRP as outlined on page 11, lines 249-251. Some of these changes were related to semantics (for instance, question 37 of PNKAS and questions 11 and 37 of PHPKASRP) and cultural changes (for instance, questions 7 and 36 of PNKAS and questions 10 and 36 of PHPKASRP).

Comment

Content validity is a relevant method especially when designing an instrument. It is not so important when the instrument has been previously validated and used in different areas. However, when an instrument is translated into another language, if the explored concepts are supposed to change significantly from one culture to another, it may be useful to recheck the face validity.

Moreover, in page 4, line 92 It is said that validity is not the property of an instrument, but depends on the interpretation related to the context and participants. However, knowledge of health is based on science, evidence, principles, theories, and is universal. Therefore, they do not depend on the interpretation of people and cultures.

Content validity evaluates qualitatively whether the questionnaire covers all the dimensions of the phenomenon to be measured, since an instrument is considered to be valid in its content if it considers all aspects related to the concept it measures. For this, it is necessary to have a clear idea of the conceptual aspects to be measured. And in this case, since the instrument had previously been validated and used in different areas, the content was already available.

Response

Thanks for the feedback. As you already know, a valid instrument is a reliable one but a reliable instrument does not guarantee validity. Thus, validity is an important quality expected of instruments. As you rightly point out and explain, there are several types of validity (content, construct, criterion, discriminant among others). The choice on the type of validity to be assessed depends on the aim of this enterprise. Our reasons for assessing content validity as the starting point for other types of validity and reliability in the future have been provided in the manuscript (on page 2, lines 25-27; page 4, lines 92-95 and page 5, lines 96-99). It is significant to also mention that, the changes which have been made in the instruments would have been missed if we had not assessed their content validity as the starting point for other types of validity and reliability assessments to be explored in the future.

Comment

Page 3; line 67. This is what I cannot understand. In this section, the questions that arise are: When talking about the revised version (PHPKASRP). Is the article review being done for the first time in this article? Or has this version already been created before? Why do you call short version if they have almost the same number of questions?...May be could you give more details or change the way to explain it.

Response

Changes have been made as found on page 4, line 71.

Comment

In the next sentence: page 3; line 68. “These two instruments were developed to measure healthcare professionals and students’ knowledge and attitudes regarding children’s pain (13)”. We find the reference of the PNKAS, what about the PHPKASRP reference? Could you explain this better?

Response

The reference for PHPKASRP has been provided on page 4, lines 72-73.

Comment

When talking about instrument validation, I would give more information about whether or not PHPKASRP is validated, etc.

Response

Information on the validity of PHPKASRP has been provided on page 4, lines 85-87.

Comment

Page 4; line 84. References for PHPKASRP are again needed.

Response

Citation of this instrument has been provided on page 4, lines 87.

Comment

The information in Tables 1 and 3 are explained in the text. Table 1 could be omitted by completing the information in the text. However, Table 3 can be kept, since it serves as a synthesis of the results.

Response

As suggested, Table 1 has been omitted whilst Table 3 has been retained as the only table within the manuscript. Table 2 has been designated as S1 Table in the supplementary file named “S1_Table.docx”.

Comment

Page 6; paragraph line 128-133.There is no specific reference to cultural and semantic factors. Despite the emphasis that has been given in the introduction by providing references.

They talk about: comprehensiveness, objectivity, organization and relevance defined as “comprehensive: that issue containing important information to reach the objective of the study, stated in a comprehensible manner; Objective: that issue which is easy to understand; organization: the disposition of the issues and alternatives as well as their content; Relevant: that question which is related to achieving the goal of the research”.

Response

Changes have been made to reflect what occurred in practice, refer to page 6, lines 130-133.

Comment

Page 8; line 182. Again, the article reference of the revised version (PHPKASRP instrument) is missed.

Response

Citation to this instrument has been provided on page 8, lines 182-183.

Comment

The results on page 10 should reference Table 3, where the results are summarized.

The authors acknowledge both the limitations and the conclusions that, as future lines, a validation of the criterion and the construct is required.

Response

References to the now “Table 1” has been provided on page 10, lines 230-231.

Comment

This type of validity requires basic statistics. Experts and / or researchers use in evaluating the relevance of a scale that requires descriptive analysis.

Response

Due to the confusion introduced by the mention of “by pediatric experts”, this phrase has been removed to enhance clarity. Refer to page 7, line 152.

---

## [Decision Letter · Decision Letter 1]

23 Jul 2020

PONE-D-20-05637R1

Evaluating the content validity of two versions of an instrument used in measuring pediatric pain knowledge and attitudes in the Ghanaian context.

PLOS ONE

Dear Dr. Kusi Amponsah,

Thank you for submitting your manuscript to PLOS ONE. After careful consideration, we feel that it has merit but does not fully meet PLOS ONE’s publication criteria as it currently stands. Therefore, we invite you to submit a revised version of the manuscript that addresses the points raised during the review process.

ACADEMIC EDITOR:

Please, put efforts to answer properly the reviewers and to improve the manuscript as suggested.

We look forward to receiving your revised manuscript.

Kind regards,

Matias Noll, Ph.D

Academic Editor

PLOS ONE

Reviewers' comments:

Reviewer's Responses to Questions

**Comments to the Author**

1. If the authors have adequately addressed your comments raised in a previous round of review and you feel that this manuscript is now acceptable for publication, you may indicate that here to bypass the “Comments to the Author” section, enter your conflict of interest statement in the “Confidential to Editor” section, and submit your "Accept" recommendation.

Reviewer #1: All comments have been addressed

Reviewer #2: (No Response)

Reviewer #3: All comments have been addressed

Reviewer #4: All comments have been addressed

Reviewer #5: (No Response)

2. Is the manuscript technically sound, and do the data support the conclusions?

Reviewer #1: Yes

Reviewer #2: Partly

Reviewer #3: Yes

Reviewer #4: Partly

Reviewer #5: Yes

3. Has the statistical analysis been performed appropriately and rigorously? 

Reviewer #1: Yes

Reviewer #2: No

Reviewer #3: Yes

Reviewer #4: Yes

Reviewer #5: Yes

4. Have the authors made all data underlying the findings in their manuscript fully available?

Reviewer #1: Yes

Reviewer #2: Yes

Reviewer #3: Yes

Reviewer #4: (No Response)

Reviewer #5: (No Response)

5. Is the manuscript presented in an intelligible fashion and written in standard English?

Reviewer #1: Yes

Reviewer #2: Yes

Reviewer #3: Yes

Reviewer #4: Yes

Reviewer #5: No

6. Review Comments to the Author

Reviewer #1: I just have to congratulate the authors for their work on modifications. They have carefully read and responded to each and every reviewer review and comment. I am very satisfied with the final work, and among all I think that the first version has improved a lot.

Reviewer #2: I thank the authors for the improvements they have made to the manuscript. However, I still have some concerns related to my original comment in the first review.

Reviewer #3: Dear Editor

Thank you again for the opportunity to review this manuscript. The manuscript aims to evaluate and compare the content validity of PNKAS and its revised version, PHPKASRP from Ghanaian perspective. The authors have done an excellent work to address the comments I raised in the first review. However, there are still some relatively minor comments that need to be addressed. Please find them below.

Introduction

Lines 70-73. “Key among them are Pediatric Nurses’ Knowledge and Attitudes Survey regarding pain (PNKAS) [13] and 71 its revised version, the Pediatric Healthcare Providers’ Knowledge and Attitudes Survey Regarding Pain (PHPKASRP) [Manworren RCB, personal communication, August 16, 2018].”

According to the authors, the developer is yet to publish the PHPKASRP. However, it has been almost two years since the developer provided the authors with the PHPKASRP. I’m wondering why the PHPKASRP was not published yet and the reasons behind that, although, according to the authors, “it has been translated into other languages and permitted for use in many organizations around the world”, and was described as a “key” instrument.

I think the PHPKASRP, until this date, does not reach the level to be a “key” instrument since there is no single publication about it, except for the abstract (ref. 17), or about its psychometric properties in English and other languages.

Lines 85-89. “Content validity of the PHPKASRP has been established by national content experts comprising of physicians, pediatric nurses and pharmacists in the United States of America [Manworren RCB, personal communication, August 16, 2018]. According to the developer, the instrument has been translated into other languages and permitted for use in many organizations around the world [Manworren RCB, personal communication, August 16, 2018].

Further, I do not think it would appropriate to state such information based on personal communication alone and not based on published evidence. The authors need to remove these statements or cite them properly.

Discussion

Lines 282-285. “In line with the recommendations of Delgado-Rico and colleagues [43], decisions on items (i.e., elimination, modification or conservation) were made on the basis of the content validity indices, feedback given by experts, inputs from the instrument developer and the contribution of the items to the overall construct under investigation”.

I still think you need to cite Delgado-Rico and colleagues in your methods since you followed their recommendations in your decisions making. The authors need to explain the reasons for not citing them if they disagree with that.

Appendices

There are several spelling mistakes in “S1 Appendix. Revised Pediatric Nurses’ Knowledge and Attitudes Survey regarding pain (r-PNKAS)”. For example, items 8, 10, 22, 23, and 33. Please correct them and check the rest of this document and other documents carefully.

Reviewer #4: The authors did a great job in adjusting the manuscript in response to the reviewers' comments, but a few issues remain. A main issue that remains for me is that the current data presented on its own isn't very strong and I still believe the current manuscript would be stronger if the authors could add data on further validation studies with the modified instruments.

Minor comments:

1) In the introduction, p. 4 line 84, it is unclear to me that the PHPKASRP is a variation of the previously discussed instrument. That only becomes clear to me on p. 8 line 180. Therefore I would suggest including this information earlier on.

2) The data analyses section does not provide any details on what is being done with the comments made by the participants on appropriateness to the culture, semantics (comprehensibility, simplicity, grammatical construction) and healthcare resources available in the Ghanaian context. It only becomes clear in the results that the purpose of these comments is to redesign these instruments. More detail on how these comments were systematically dealt with to redesign the instruments is needed.

3) I find it strange that the development of the PNKAS and PHPKASRP instruments is detailed at the end of the methods sections, I would prefer to see this more upfront, might even fit in the introduction.

Reviewer #5: Dear authors,

I highly recommend you to discuss and take in account previous articles that purpose questionnaires for evaluation of Back Pain. In a quick search I fond some articles that can be improve your introduction as well as the discussion, as follow:

Spanish translation, cross-cultural adaptation and validation of the Argentine version of the Back Pain Attitudes Questionnaire Pierobon, A., Policastro, P.O., Soliño, S., (...), Raguzzi, I.A., Villalba, F.J. 2020 Musculoskeletal Science and Practice

46,102125

Is There Equivalence between the Electronic and Paper Version of the Questionnaires for Assessment of Patients with Chronic Low Back Pain? Azevedo, B.R., Oliveira, C.B., Araujo, G.M.D., (...), Pinto, R.Z., Christofaro, D.G.D. 2020 Spine

45(6), pp. E329-E335

Back Pain and Body Posture Evaluation Instrument (BackPEI): Development, content validation and reproducibility Noll, M., Tarragô Candotti, C., Vieira, A., Fagundes Loss, J. 2013 International Journal of Public Health

58(4), pp. 565-572

Psychometric Study and Content Validity of a Questionnaire to Assess Back-Health-Related Postural Habits in Daily Activities Monfort-Pañego, M., Miñana-Signes, V. 2020 Measurement in Physical Education and Exercise Science

Validation of the Japanese Version of the Fremantle Back Awareness Questionnaire in Patients with Low Back Pain Nishigami, T., Mibu, A., Tanaka, K., (...), Stanton, T.R., Moseley, G.L. 2018 Pain Practice

18(2), pp. 170-179

7. PLOS authors have the option to publish the peer review history of their article (what does this mean?). If published, this will include your full peer review and any attached files.

Reviewer #1: **Yes: **Vicente Miñana-Signes (PhD)

Body Languages Didactics Department

Teacher Training Faculty

University of Valencia

Reviewer #2: No

Reviewer #3: No

Reviewer #4: No

Reviewer #5: No

---

## [Author Response · Author response to Decision Letter 1]

1 Oct 2020

Overall Comments from Reviewer 1

Reviewer #1: I just have to congratulate the authors for their work on modifications. They have carefully read and responded to each and every reviewer review and comment. I am very satisfied with the final work, and among all I think that the first version has improved a lot.

Response

Thank you.

Overall Comments from Reviewer 2

Reviewer #2: I thank the authors for the improvements they have made to the manuscript. However, I still have some concerns related to my original comment in the first review.

Response 

Thanks for accepting to review our manuscript. Responses to the additional concerns have been provided below.

Overall Comments from Reviewer 3

Reviewer #3: Dear Editor

Thank you again for the opportunity to review this manuscript. The manuscript aims to evaluate and compare the content validity of PNKAS and its revised version, PHPKASRP from Ghanaian perspective. The authors have done an excellent work to address the comments I raised in the first review. However, there are still some relatively minor comments that need to be addressed. Please find them below.

Response

Thank you all for the feedback. Please find below our responses to the concerns raised. 

Specific Comments from Reviewers 1-3

Comment

Introduction

Lines 70-73. “Key among them are Pediatric Nurses’ Knowledge and Attitudes Survey regarding pain (PNKAS) [13] and 71 its revised version, the Pediatric Healthcare Providers’ Knowledge and Attitudes Survey Regarding Pain (PHPKASRP) [Manworren RCB, personal communication, August 16, 2018].”

According to the authors, the developer is yet to publish the PHPKASRP. However, it has been almost two years since the developer provided the authors with the PHPKASRP. I’m wondering why the PHPKASRP was not published yet and the reasons behind that, although, according to the authors, “it has been translated into other languages and permitted for use in many organizations around the world”, and was described as a “key” instrument.

I think the PHPKASRP, until this date, does not reach the level to be a “key” instrument since there is no single publication about it, except for the abstract (ref. 17), or about its psychometric properties in English and other languages.

Response

Thanks for the feedback. We do not know why the instrument developer (Manworren, R.C.B) has not yet published findings from the PHPKASRP. She however, indicated in one of our email correspondences that our work has served as a reminder for her to publish findings related to the PHPKASRP. We have amended the statement relating to the PHPKASRP as a “key” instrument on the subject. Refer to page 3, lines 66-71.

Comment

Lines 85-89. “Content validity of the PHPKASRP has been established by national content experts comprising of physicians, pediatric nurses and pharmacists in the United States of America [Manworren RCB, personal communication, August 16, 2018]. According to the developer, the instrument has been translated into other languages and permitted for use in many organizations around the world [Manworren RCB, personal communication, August 16, 2018].

Further, I do not think it would appropriate to state such information based on personal communication alone and not based on published evidence. The authors need to remove these statements or cite them properly.

Response

Amendments have been made to reflect the state of affairs regarding these issues. Refer to page 4, lines 85-88.

Comment

Discussion

Lines 282-285. “In line with the recommendations of Delgado-Rico and colleagues [43], decisions on items (i.e., elimination, modification or conservation) were made on the basis of the content validity indices, feedback given by experts, inputs from the instrument developer and the contribution of the items to the overall construct under investigation”.

I still think you need to cite Delgado-Rico and colleagues in your methods since you followed their recommendations in your decisions making. The authors need to explain the reasons for not citing them if they disagree with that.

Response

As suggested, Delgado-Rico and colleagues have been cited under the materials and methods section on page 8, lines 173-176.

Comment

Appendices

There are several spelling mistakes in “S1 Appendix. Revised Pediatric Nurses’ Knowledge and Attitudes Survey regarding pain (r-PNKAS)”. For example, items 8, 10, 22, 23, and 33. Please correct them and check the rest of this document and other documents carefully.

Response

Spelling errors have been corrected in the “S1_Appendix” and “S2_Appendix” in the revised documents. 

Overall Comments from Reviewer 4

Reviewer #4: The authors did a great job in adjusting the manuscript in response to the reviewers' comments, but a few issues remain. A main issue that remains for me is that the current data presented on its own isn't very strong and I still believe the current manuscript would be stronger if the authors could add data on further validation studies with the modified instruments.

Response

Thanks for the feedback. We humbly disagree with the comments relating to the strength of the current manuscript due to the absence of other validation studies as there is no published recommendation on the number of validation studies to be included in a manuscript to qualify for the supposed strength. This notwithstanding, we acknowledge the importance of additional psychometric testing (validity and reliability) of the modified instruments which have been stated in the discussion (on page 33, lines 345-347) and conclusion (on page 33, lines 356-357) sections of our manuscript. As indicated in the recommendations, other types of validity and reliability of the modified instrument would be pursued in the future. 

Specific Comments from Reviewers 

Comment

In the introduction, p. 4 line 84, it is unclear to me that the PHPKASRP is a variation of the previously discussed instrument. That only becomes clear to me on p. 8 line 180. Therefore, I would suggest including this information earlier on.

Response

Information on the above have been provided earlier on at page 3, lines 67-71.

Comment

The data analyses section does not provide any details on what is being done with the comments made by the participants on appropriateness to the culture, semantics (comprehensibility, simplicity, grammatical construction) and healthcare resources available in the Ghanaian context. It only becomes clear in the results that the purpose of these comments is to redesign these instruments. More detail on how these comments were systematically dealt with to redesign the instruments is needed.

Response

Details on the application of the comments in redesigning the instruments have been provided on page 8, lines 183-189.

Comment

I find it strange that the development of the PNKAS and PHPKASRP instruments is detailed at the end of the methods sections, I would prefer to see this more upfront, might even fit in the introduction.

Response

As suggested, information on the instrument development processes has been moved upfront. Refer to page 5, lines 99-120 and page 6, lines 121-132.

Overall Comments from Reviewer 5

Comment

I highly recommend you to discuss and take in account previous articles that purpose questionnaires for evaluation of Back Pain. In a quick search I found some articles that can be improve your introduction as well as the discussion, as follow:

Spanish translation, cross-cultural adaptation and validation of the Argentine version of the Back Pain Attitudes Questionnaire Pierobon, A., Policastro, P.O., Soliño, S., (...), Raguzzi, I.A., Villalba, F.J. 2020 Musculoskeletal Science and Practice

46,102125

Is There Equivalence between the Electronic and Paper Version of the Questionnaires for Assessment of Patients with Chronic Low Back Pain? Azevedo, B.R., Oliveira, C.B., Araujo, G.M.D., (...), Pinto, R.Z., Christofaro, D.G.D. 2020 Spine

45(6), pp. E329-E335

Back Pain and Body Posture Evaluation Instrument (BackPEI): Development, content validation and reproducibility Noll, M., Tarragô Candotti, C., Vieira, A., Fagundes Loss, J. 2013 International Journal of Public Health

58(4), pp. 565-572

Psychometric Study and Content Validity of a Questionnaire to Assess Back-Health-Related Postural Habits in Daily Activities Monfort-Pañego, M., Miñana-Signes, V. 2020 Measurement in Physical Education and Exercise Science

Validation of the Japanese Version of the Fremantle Back Awareness Questionnaire in Patients with Low Back Pain Nishigami, T., Mibu, A., Tanaka, K., (...), Stanton, T.R., Moseley, G.L. 2018 Pain Practice

18(2), pp. 170-179

Response

Upon reading the suggested articles, some clarifications have been made in the discussion section. Refer to page 31, lines 297-299, 311-313.

---

## [Decision Letter · Decision Letter 2]

26 Oct 2020

Evaluating the content validity of two versions of an instrument used in measuring pediatric pain knowledge and attitudes in the Ghanaian context.

PONE-D-20-05637R2

Dear Dr. Kusi Amponsah,

We’re pleased to inform you that your manuscript has been judged scientifically suitable for publication and will be formally accepted for publication once it meets all outstanding technical requirements.

Kind regards,

Matias Noll, Ph.D

Academic Editor

PLOS ONE

Additional Editor Comments (optional):

Reviewers' comments:

Reviewer's Responses to Questions

**Comments to the Author**

1. If the authors have adequately addressed your comments raised in a previous round of review and you feel that this manuscript is now acceptable for publication, you may indicate that here to bypass the “Comments to the Author” section, enter your conflict of interest statement in the “Confidential to Editor” section, and submit your "Accept" recommendation.

Reviewer #3: All comments have been addressed

2. Is the manuscript technically sound, and do the data support the conclusions?

Reviewer #3: Yes

3. Has the statistical analysis been performed appropriately and rigorously? 

Reviewer #3: Yes

4. Have the authors made all data underlying the findings in their manuscript fully available?

Reviewer #3: Yes

5. Is the manuscript presented in an intelligible fashion and written in standard English?

Reviewer #3: Yes

6. Review Comments to the Author

Reviewer #3: After reading the revised manuscript, the authors have addressed all the comments I have raised. I would like to thank them for addressing my comments.

7. PLOS authors have the option to publish the peer review history of their article (what does this mean?). If published, this will include your full peer review and any attached files.

Reviewer #3: **Yes: **Hamad S. Al Amer, PT, PhD

---

## [Editor Report · Acceptance letter]

28 Oct 2020

PONE-D-20-05637R2 

Evaluating the content validity of two versions of an instrument used in measuring pediatric pain knowledge and attitudes in the Ghanaian context 

Dear Dr. Kusi Amponsah:

I'm pleased to inform you that your manuscript has been deemed suitable for publication in PLOS ONE. Congratulations! Your manuscript is now with our production department. 

Kind regards, 

on behalf of

Dr. Matias Noll 

Academic Editor

PLOS ONE